# Ubiquitination of stalled ribosomes enables mRNA decay via HBS-1 and NONU-1 *in vivo*

**Parissa C. Monem[ID], Nitin Vidyasagar[ID], Audrey L. Piatt, Enisha Sehgal, Joshua A. Arribere[ID]***

Department of Molecular, Cell, and Developmental Biology, University of California at Santa Cruz, Santa Cruz, California, United States of America

* jarriber@ucsc.edu

## Abstract

As ribosomes translate the genetic code, they can encounter a variety of obstacles that hinder their progress. If ribosomes stall for prolonged times, cells suffer due to the loss of translating ribosomes and the accumulation of aberrant protein products. Thus to protect cells, stalled ribosomes experience a series of reactions to relieve the stall and degrade the offending mRNA, a process known as No-Go mRNA Decay (NGD). While much of the machinery for NGD is known, the precise ordering of events and factors along this pathway has not been tested. Here, we deploy *C. elegans* to unravel the coordinated events comprising NGD. Utilizing a novel reporter and forward and reverse genetics, we identify the machinery required for NGD. Our subsequent molecular analyses define a functional requirement for ubiquitination on at least two ribosomal proteins (eS10 and uS10), and we show that ribosomes lacking ubiquitination sites on eS10 and uS10 fail to perform NGD *in vivo*. We show that the nuclease NONU-1 acts after the ubiquitin ligase ZNF-598, and discover a novel requirement for the ribosome rescue factors HBS-1/PELO-1 in mRNA decay via NONU-1. Taken together, our work demonstrates mechanisms by which ribosomes signal to effectors of mRNA repression, and we delineate links between repressive factors working toward a well-defined NGD pathway.

## Author summary

Ribosomes are large molecular machines entrusted with the essential task of reading mRNAs and building proteins. It is critical that ribosomes proceed with their work undeterred, so as to avoid traffic jams between ribosomes on mRNAs; however, they sometimes experience challenges which make traffic jams unavoidable. In these cases, cells have evolved machinery to clean up stalled ribosomes and remove offending mRNAs. Much of this machinery is now identified, but the relationships between factors has yet to be tested. Here, in *C. elegans* we identify the machinery that responds to stalls, and we order factors relative to one another. We find that ribosomes must be tagged by ubiquitin to signal mRNA decay, as ribosomes lacking ubiquitin sites fail to elicit mRNA decay. Furthermore, we find that these ubiquitin signals recruit an enzyme which cuts mRNA, and we present evidence that ribosome rescue factors are required for the mRNA decay

**Data Availability Statement:** The data discussed in this publication are accessible through GEO Series accession number GSE218514 and S2 Table.

**Funding:** This work was supported in part by a T32 Training Grant Fellowship from the National

Institute of General Medical Sciences (NIGMS) (5T32GM133391) to P.C.M., an R01 grant from the NIGMS (1R01GM131012) to J.A.A., a Searle Scholars award to J.A.A., and start-up funds from UCSC to J.A.A. The funders did not play any role in the study design, data collection and analysis, decision to publish, or preparation of the manuscript.

**Competing interests:** The authors have declared that no competing interests exist.

reaction as well. Overall, our work provides new connections between factors orchestrating the elimination of problematic mRNAs that stall ribosomes.

## Introduction

An organism's growth, function, and response to environmental changes rely on ribosomes accurately producing proteins. Subtle errors in mRNAs are often elusive, requiring active translation to detect and trigger downstream repression. If left unchecked, defective mRNAs have the potential to produce toxic proteins resulting in disease phenotypes, such as neurodegeneration [1,2].

Ribosomes that translate problematic mRNAs can elicit various repressive mechanisms. Two such mechanisms are Nonstop mRNA Decay (NSD) and No-Go mRNA Decay (NGD). NSD is triggered by mRNAs lacking stop codons, which can arise from polyadenylation or from mRNA cleavage [3]. NGD occurs on mRNAs containing a variety of elongation-inhibiting features, such as stable secondary structures, stretches of rare codons, polybasic amino acid-encoding sequences, or damaged nucleotides [4]. While triggered by different mRNA species, both of these pathways result in mRNA decay, ribosome rescue, and nascent peptide degradation through the coordinated recruitment of several effectors.

One key event in NGD is the generation of a stalled, collided ribosome species at a problematic site on the mRNA. This species is thought to be unique to NGD and could conceivably recruit downstream effectors [5,6,7]. The interface between collided ribosomes is the site of ubiquitination events on multiple ribosomal proteins near the mRNA's path [6,7]. The conserved E3 ubiquitin ligase ZNF-598 is thought to deposit ubiquitin marks on at least two ribosomal proteins, including RPS-10 (eS10) and RPS-20 (uS10) [8,9]. Prior work overexpressing ubiquitination-deficient point mutations of eS10 suggests that ribosomal protein ubiquitination is important for NGD [8], though whether additional, ubiquitination-independent mechanisms contribute remains unclear. Our limited understanding of ribosomal ubiquitination is in part due to the difficulty in recovering viable mutants of the target ribosomal proteins, which has thus far complicated a straightforward analysis of individual ubiquitination sites and their relationship to repression via ZNF-598. Harnessing a system to study the functional contributions of ribosomal ubiquitination would clarify its importance.

Early NGD work pointed towards *Saccharomyces cerevisiae* Dom34 and Hbs1 (Pelota and HBS1 in higher eukaryotes) as being important for mRNA cleavage [4]. Despite an early study claiming nuclease activity of Dom34, it is now thought that this requirement is indirect [10] with subsequent work on Dom34/Hbs1 showing a biochemical role in ribosome rescue [11,12,13,14]. Later work in both *Caenorhabditis elegans* and *S. cerevisiae* identified NONU-1/Cue2/YPL199C as endonucleases responsible for cleaving targets of NSD and NGD in the vicinity of stalled ribosomes [15,16]. While NONU-1 is required for efficient NSD and NGD, it remains unclear how and when the factor is targeted to stall-inducing mRNAs. However, clues to its recruitment mechanism exist: NONU-1 and its homologs contain at least two conserved ubiquitin-binding CUE domains [15].

Much of what is known about NGD comes from studying the effect of individual factors and events, and it is unclear how these steps relate to one another to bring about target mRNA repression. Here, we deployed *C. elegans* to unravel the series of events during NGD. We show that mutation of ribosomal ubiquitination sites on RPS-10 (eS10) and RPS-20 (uS10) phenocopies knock out of ZNF-598. We present data in support of a model in which ZNF-598 first ubiquitinates ribosomes at stall sites, followed by mRNA degradation via NONU-1.

Interestingly, we also recovered a role for HBS-1 and PELO-1 in mRNA decay via NONU-1 cleavage, consistent with early northern data in *S. cerevisiae*, suggesting that ribosome rescue may be an important step that precedes mRNA cleavage.

## Results

### A novel ribosome stalling screen identifies core NGD machinery

To establish *C. elegans* as a system to study NGD, we built a ribosome stalling reporter using CRISPR/Cas9 at the *unc-54* locus. UNC-54 encodes a major muscle myosin required for animal movement, but dispensable for life [17,18,19]. UNC-54 has also been the starting point for a number of genetic screens, and its expression is relatively well-characterized. To build a NGD reporter, we added a T2A 'stop-and-go' peptide, a FLAG tag, twelve rare arginine codons, and GFP to the C-terminus of UNC-54 via CRISPR/Cas9, and we call the resultant allele *unc-54(rareArg)* (Fig 1A) (reporter sequence available in S1 Table). We selected rare arginine codons for two reasons: (1) arginine is a positively charged amino acid and prior work [20,21] suggests this may induce ribosome stalling via interactions with the peptide exit tunnel, and (2) the Arg-tRNAs decoding two of the rarest codons (CGG, AGG) in *C. elegans* are very lowly expressed under normal growth conditions (Aidan Manning, Todd Lowe, personal communication, June 2021), providing a second avenue by which stalling may occur.

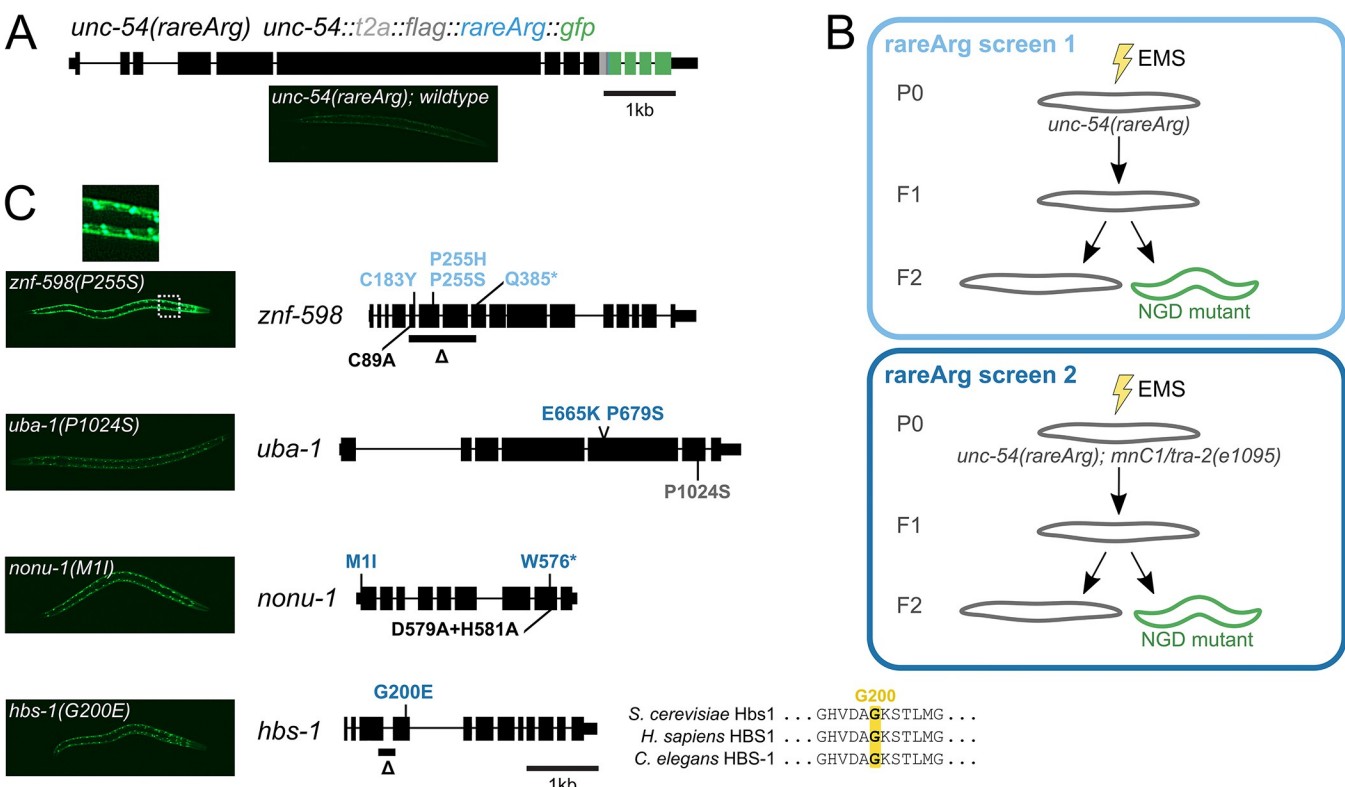

**Fig 1. Genetic screens identify suppressors of No-Go mRNA Decay.** **(A)** Gene diagram showing annotated exons (black rectangles) of *unc-54(rareArg)*. Colored rectangles represent CRISPR/Cas9 insertions at the endogenous *unc-54* locus: T2A sequence (gray), FLAG (dark gray), 12 rare arginine codons (blue), and GFP (green). **(B)** Schematics of rareArg genetic screens. **(C)** *znf-598*, *uba-1*, *nonu-1*, and *hbs-1* alleles with representative image of one allele per gene on the left. Black rectangles represent exons, thicker rectangles are CDS, and thin lines are introns. Mutations made via EMS in the rareArg screen 1 (light blue) or rareArg screen 2 (dark blue), and via CRISPR/Cas9 (black) or CGC (gray) are shown. For HBS-1, multiple sequence alignment shows conserved glycine (G200) in GTPase domain.

Strains with the *unc-54(rareArg)* construct showed an uncoordinated (Unc) phenotype consistent with reduction of UNC-54 protein. As other C-terminal tags of UNC-54 are functional [22], the loss of UNC-54 protein suggests that mRNAs produced from the *unc-54(rareArg)* locus are repressed. *unc-54(rareArg)* animals also exhibited very low levels of GFP, suggesting that few ribosomes make it to the 3'end of *unc-54(rareArg)* due to reduced mRNA levels and/or high amounts of stalling during translation elongation.

To initially validate the *unc-54(rareArg)* reporter as a target of NGD, we crossed it with alleles of two factors known to be required for NGD in *C. elegans* (*nonu-1(AxA)*) and other systems (*znf-598(Δ)*) [8,9,15,23]. In each case we observed de-repression of the reporter, manifest as increased fluorescence and an improvement in animal movement and egg laying (UNC-54 is required in the vulva muscles for egg laying). The *nonu-1* result is consistent with our prior work showing that this stretch of twelve rare arginine codons confers *nonu-1*-dependent mRNA decay [15]. Notably, the phenotypic effects seen upon *nonu-1* knockout differ from that seen in *S. cerevisiae*: knockout of the homologous *CUE2* in *S. cerevisiae* only confers effects upon simultaneous knockout of additional factors [16]. Given that most of our mechanistic understanding of NGD comes from work in *S. cerevisiae* [4,5,16], and that genetic screens in human K562 cells failed to identify the NONU-1 homolog [24], we reasoned that a genetic screen in *C. elegans* would prove insightful and augment information gained from other systems.

Using *unc-54(rareArg)*, we performed two genetic screens (Methods) (Fig 1B). In the first of these screens, we EMS-treated *unc-54(rareArg)* animals, harvested eggs, and screened ~90,000 F1 genomes. Among these, we found six mutants across six plates with increased movement (indicative of de-repression of *unc-54(rareArg)*) and increased GFP (indicative of translation to the end of the *unc-54(rareArg)* transcript). Evidence of stall readthrough was visible by the nuclear localization of GFP upon its de-repression (Fig 1C). This observation revealed our serendipitous construction of an N-terminal motif (a run of arginines) matching the sequence requirements for a nuclear localization signal [25]. We genetically mapped and identified variants, and found that four alleles mapped to *znf-598* (Fig 1C).

Reasoning that there was more to NGD than *znf-598*, we repeated the screen using a chromosomal balancer covering the *znf-598* locus to preclude recovery of recessive *znf-598* mutations. We screened an additional ~105,000 F1 genomes, among which we found an additional 14 mutants across 14 plates. The increase of movement and GFP in these animals were less pronounced than the *znf-598* mutants from the first screen. Nevertheless, the phenotypes were robust enough to allow us to genetically map and identify causative loci. Here we report a total of nine alleles in four genes from both screens (Fig 1C). Five alleles in one additional gene were identified (*catp-6*, S1 Table); mapping and characterization of the remaining six mutants is ongoing.

Seven alleles were split amongst three readily-identifiable NGD components: *nonu-1*, *znf-598*, and *hbs-1*. We found two mutations in *nonu-1*, as would be expected based on the *unc-54(rareArg)* reporter validation. We isolated four mutations in *znf-598* (all from the first genetic screen), three of which were missense mutations. One mutation (C183Y) was within a region that matches the consensus for a C2H2 zinc finger in *C. elegans* and *Homo sapiens*, but is not conserved in *S. cerevisiae*. Two alleles at P255 (P255S, P255H) mutated a highly conserved proline at the start of a C2H2 zinc finger. One allele was found in the *C. elegans*' homolog of HBS1 (*k07a12.4*, hereafter *hbs-1*). This allele was in a conserved GTP-binding and active site residue (G200E) (Fig 1C), consistent with a functional requirement for GTP binding and hydrolysis by HBS-1 in NGD.

Two additional mutations mapped in *uba-1*, the sole E1 ubiquitin-activating enzyme in *C. elegans*. Both *uba-1* mutations (E665K, P679S) exhibited poor viability, as would be expected

as complete loss of *uba-1* is thought to be lethal [26]. We confirmed the *uba-1* result by crossing in a known temperature-sensitive loss-of-function allele of this gene (*it129*; *P1024S*), and observed temperature-dependent de-repression of GFP. As ZNF-598 is known to function in other systems as a ubiquitin ligase [8,9], we expect that *uba-1* loss-of-function compromised the ubiquitin-conjugation cascade, giving rise to *unc-54(rareArg)* de-repression.

## Cell-specific NGD rescue via overexpression of factors

We also developed a system to analyze the effects of overexpression of particular NGD pathway components using extrachromosomal arrays. Overexpression is a useful means to generate hyperactive alleles of factors to test their relative order in a pathway. Injection of foreign DNA into the germline of *C. elegans* results in fusion of the DNA sequences into an extrachromosomal array [27,28]. The resultant array can be both meiotically and mitotically inherited, albeit with reduced efficiencies compared to endogenous chromosomes. We created a plasmid vector to overexpress a gene of interest on a transgenic array with the following components: (1) a *myo-3* promoter, to drive expression in the body wall muscle where *unc-54* is also expressed, (2) a fluorescent mCherry protein, to monitor inheritance and expression of the array, (3) a 'stop-and-go' T2A sequence, to separate mCherry from the gene-of-interest, and (4) a site to insert a gene-of-interest. We constructed plasmids encoding fluorescent proteins linked to *C. elegans'* *znf-598* and *nonu-1*.

To check the functionality of factors expressed from an array, we made overexpression arrays in each of the cognate mutant backgrounds. Overexpression of the wild-type ZNF-598 and NONU-1 protein restored repression of *unc-54(rareArg)* and rescued the loss-of-function phenotype of *znf-598* (Fig 2A) and *nonu-1* (Fig 2B), respectively. Thus the overexpressed factors were functional. Arrays can be stochastically lost or silenced in cell lineages of the animal, allowing us to examine rescue on a cell-by-cell level. For each of *znf-598* and *nonu-1*, we observed an inverse relationship between factor expression (monitored via mCherry) and *unc-54(rareArg)* (monitored via GFP). Thus the rescue was cell-autonomous, as would be expected under current models of ZNF-598 and NONU-1 acting directly on ribosomes and mRNAs.

To quantify the inverse relationship of factor overexpression (mCherry) to NGD (GFP), we calculated an overlap score, based on the brightest red and green pixels across multiple, independent animals (Methods) (S1 Fig). If mCherry and GFP are non-overlapping (as would be expected from rescue), the overlap score would be negative. If mCherry and GFP overlap (as would be expected without rescue), the overlap score would be positive. For *znf-598* and *nonu-1*, we observed values close to -1 (Fig 2C), demonstrating the generality of rescue across animals.

With a functional readout of NGD (*unc-54(rareArg)*), mutants in factors (*znf-598*, *nonu-1*, *hbs-1*, and *pelo-1*), and the overexpression/rescue system, we set out to characterize the molecular mechanisms by which factors relate and repress gene expression in response to ribosomal stalling.

## ZNF-598 is required for ribosomal ubiquitination in *C. elegans*

In the NGD screen, we recovered several alleles of *znf-598*. ZNF-598 homologs are thought to recognize ribosomal collisions and ubiquitinate sites on the small subunit, including RPS-10 (eS10) and RPS-20 (uS10) [8,9]. By multiple sequence alignment, we identified a highly conserved cysteine (C89) (Fig 3A) [9] known to be required for ubiquitin conjugation by Hel2/ ZNF598, and we generated a point mutation (C89A) via CRISPR/Cas9 at the endogenous locus. The *znf-598(C89A)* mutant displayed *unc-54(rareArg)* de-repression indistinguishable

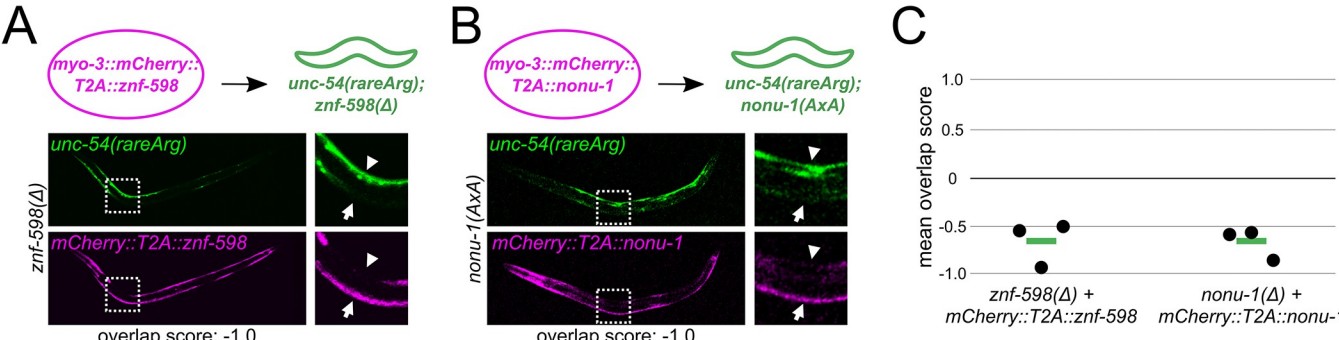

**Fig 2. : Cell-specific NGD rescue via overexpression of factors. (A)** Schematic of *znf-598* construct plasmid and *znf-598* strain subject to germline microinjection. Below are GFP and mCherry images of a representative animal expressing the above construct, with a zoom in of an area demonstrating the effect of mCherry-marked factor expression on NGD (GFP). **(B)** As in (A) for *nonu-1* construct in *nonu-1* strain. **(C)** Mean overlap score of strains in (A, B). Each black dot represents the mean of one independent isolate (n≥4 animals/isolate), with the mean of all isolates shown as a green bar.

from the phenotype of *znf-598(Δ)* (Fig 3B), supporting a role for ubiquitination in repression of stall-inducing mRNAs in *C. elegans*, as in other systems.

To investigate the ribosomal protein targets of ZNF-598, we tagged both RPS-10 (eS10) and RPS-20 (uS10) at the endogenous locus via CRISPR/Cas9. By immunoblot we observed robust expression of the cognate ribosomal protein in each strain (Fig 3C). We also mutated

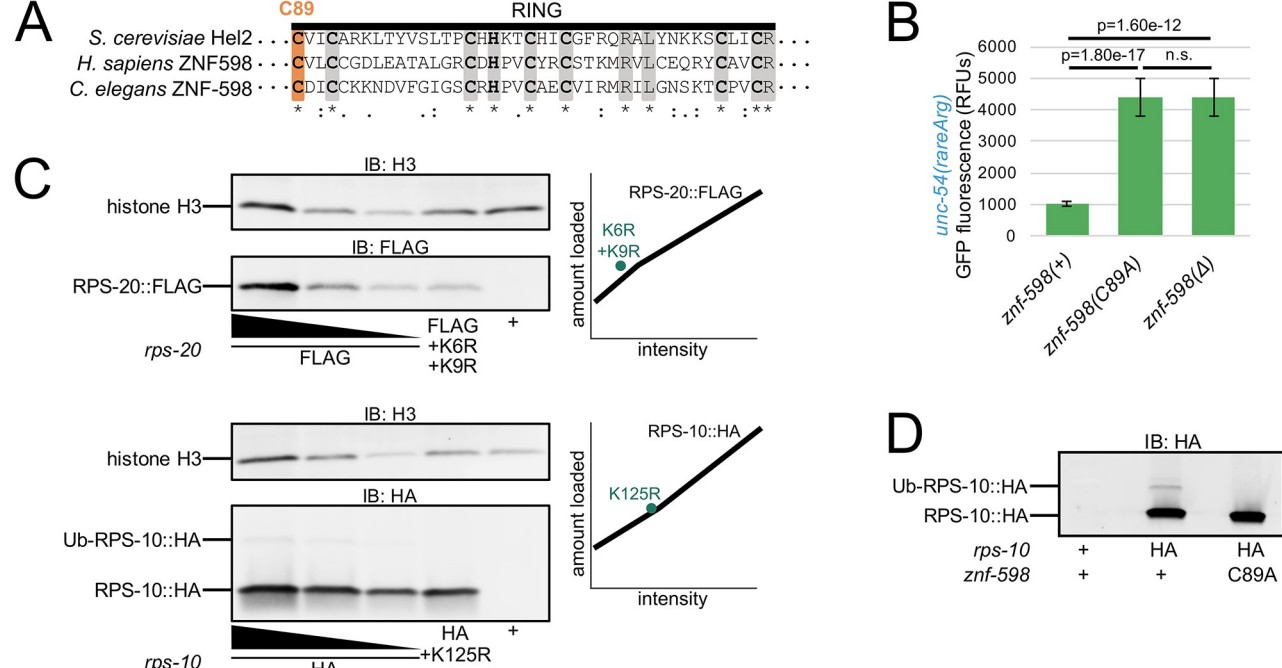

**Fig 3. ZNF-598 is required for ribosomal ubiquitination in *C. elegans*. (A)** Multiple sequence alignment of *S. cerevisiae* Hel2, *H. sapiens* ZNF598, and *C. elegans* ZNF-598 RING finger domains. Conserved residues (gray) and C89 (orange) are highlighted. Conservation below alignment is as follows: asterisks indicate identity, colons indicate amino acids with strongly similar properties, periods indicate amino acids with weakly similar properties. **(B)** Mean RFUs (relative fluorescence units) of indicated strains (n≥15 animals/strain) in the *unc-54(rareArg)* background. One standard deviation shown as error bars. p values from Welch's t-test. **(C)** Western blot of indicated strains to monitor RPS-20 and RPS-10 expression. Dilutions of wild type tagged proteins were loaded as indicated, with two-fold more and two-fold less than other two samples, to generate a standard curve shown as a black line in plots on the right. Lysine mutants of tagged proteins were quantified and plotted as teal points in the plots on the right. **(D)** Western blot of indicated strains to monitor ubiquitination of HA-tagged RPS-10.

conserved lysines in RPS-10 and RPS-20 known to be sites of ubiquitination in other systems (S2A and S2B Fig) [8,9]. The K>R substitutions did not adversely impact RPS-10 and RPS-20 expression (Fig 3C) nor animal viability. In the case of RPS-10, we also observed a band corresponding to the expected size of Ub-RPS-10. This band was absent in *rps-10(K125R)* (Fig 3C) and *znf-598(C89A)* mutants (Fig 3D). These results show that *znf-598* is required for ubiquitination of RPS-10 on K125.

## NGD-deficient ribosomes made via ablation of ubiquitination sites

Prior work underscored the importance of ribosomal ubiquitination events in NGD [8,9]. However, the precise functional contributions of individual ubiquitination sites has remained unclear, in part due to the difficulty in obtaining viable mutants in the relevant ribosomal proteins. In prior work, overexpression of RPS-10 with K>R substitutions at ubiquitination sites conferred de-repression of a stalling reporter in between that observed in wild-type and ZNF598 knockout human cells [8]. Interpretation of this experiment is complicated by a number of factors, including residual expression of wild-type RPS-10. Would complete removal of RPS-10 ubiquitination sites mimic loss of ZNF598? Does ZNF598 contribute to functional repression outside of its role in ribosomal ubiquitination? Our work thus far supported a NGD mechanism similar to that of human cells, and therefore our system seemed a useful model to explore these questions. In particular, the viability of ribosomal point substitutions at endogenous loci as well as the ease of making double mutants and overexpression constructs provided us with the means to test models of the functional importance of ribosomal ubiquitination and its relationship to ZNF-598.

To initially test for a functional role of ribosomal ubiquitination in NGD, we crossed *rps-10 (K125R)* and *rps-20(K6R+K9R)* into *unc-54(rareArg)*. In each case we observed a defect in NGD, manifest as an increase in GFP produced by *unc-54(rareArg)* (Fig 4A). The double mutant (*rps-10(K125R)*; *rps-20(K6R+K9R)*) exhibited an even stronger defect in NGD, comparable to that observed in *znf-598* mutants. We interpret these data to indicate that ribosomal ubiquitination is required for NGD, and that ablation of these two sites alone is sufficient to prevent functional NGD. Interestingly, these experiments demonstrate that it is possible to make a version of the ribosome deficient in NGD via mutation of a few lysines, highlighting the central role of the ribosome and ubiquitination in NGD.

We also performed additional experiments to clarify the function of ribosomal ubiquitination in relation to *znf-598*. First, we performed a double mutant analysis. In a model where ribosomal ubiquitination is functionally independent of *znf-598*, we would expect that a *rps-10 (K125R)*; *znf-598(Δ)* mutant would exhibit a stronger NGD phenotype (more GFP) than either single mutant alone. However, if ZNF-598 and Ub-RPS-10 lie on the same pathway, we would expect the *rps-10(K125R)*; *znf-598(Δ)* phenotype to resemble one of the single mutants. The *rps-10(K125R)*; *znf-598(Δ)* phenotype was indistinguishable from the *znf-598(Δ)* single mutant (Fig 4A), consistent with the latter model. Second, we used our overexpression system. In a model where *rps-10(K125R)* and *rps-20(K6R+K9R)* are on a partially redundant pathway to *znf-598* (*i.e.*, that *znf-598* contains additional repressive functions outside of ubiquitination at these two sites), we would expect that overexpression of ZNF-598 would provide restoration of NGD in the *rps-10(K125R)*; *rps-20(K6R+K9R)* mutant. However, if ZNF-598 acts through RPS-10 and RPS-20, overexpression of ZNF-598 in a *rps-10(K125R)*; *rps-20(K6R+K9R)* mutant would yield the same phenotype as *rps-10(K125R)*; *rps-20(K6R+K9R)*. We observed the latter (Fig 4B), again suggesting that ZNF-598 works through ubiquitination of RPS-10 and RPS-20.

Taken together, our experiments suggest that ZNF-598 is a ubiquitin ligase required for ubiquitination of at least RPS-10 and RPS-20, and that these ubiquitination events are essential

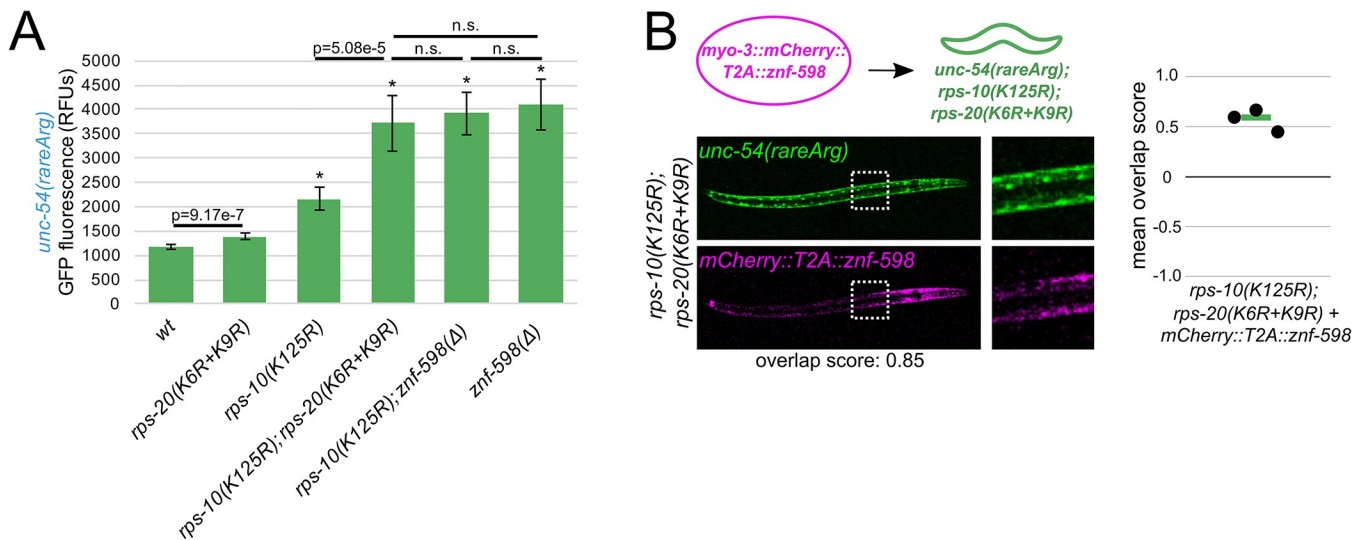

**Fig 4. NGD-deficient ribosomes made via ablation of ubiquitination sites.** (A) Mean RFUs (relative fluorescence units) of indicated strains (n≥15 animals/strain) in the *unc-54(rareArg)* background. One standard deviation shown as error bars. p values from Welch's t-test, with asterisks indicating p<0.01 for all comparisons with wild type. **(B)** As in Fig 2, with *znf-598* construct in *rps-10; rps-20* strain.

to NGD. The fact that loss of *znf-598* can be mimicked by mutation of ribosomal proteins provides compelling evidence that the primary function of ZNF-598 during NGD is to deposit ubiquitins on ribosomes.

## NONU-1 function during NGD requires CUE domains and follows ZNF-598

Having established a requirement for ubiquitination by ZNF-598 in this system, we decided to investigate the relationship of ubiquitination to mRNA decay. A key effector of NGD is NONU-1, which was identified in our screen (Fig 1C) and also in our prior NSD screen [15]. In our prior work, we noted that NONU-1 and its homologs contain CUE domains (Fig 5A) [15,29], which are known to bind ubiquitin, suggesting a mechanism of recruitment for NONU-1 to sites of ribosome stalling. To test a requirement for the CUE domains in *nonu-1* function, we deleted the CUE domains and observed a phenotype indistinguishable from other *nonu-1* mutants (Fig 5B). This result is consistent with CUE domains being essential to NONU-1 function. We also attempted to examine expression of NONU-1 by tagging the N-terminus (S1 Table), but tagged alleles were non-functional. We did not tag the C-terminus of NONU-1 as it is conserved. We look forward to future work where we can determine whether the requirement of CUE domains in NONU-1 is merely for NONU-1 expression or for its biochemical functions.

To determine whether NONU-1 acts in the same pathway as ZNF-598, we performed a double mutant analysis. If the two factors function in different pathways, we would expect additive phenotypes on the *unc-54(rareArg)* reporter in a double mutant. However, if NONU-1 and ZNF-598 work in the same pathway of repression, we would expect the double mutant to resemble one of the single mutants. We combined mutations in *znf-598* and ribosomal ubiquitination sites with *nonu-1(AxA)*, a catalytic mutation of *nonu-1* that is indistinguishable from a deletion of *nonu-1* (S3 Fig) [15]. The *nonu-1; znf-598* double mutant mimicked the *znf-598* single mutant (Fig 5C), consistent with the two factors functioning in the same pathway in NGD.

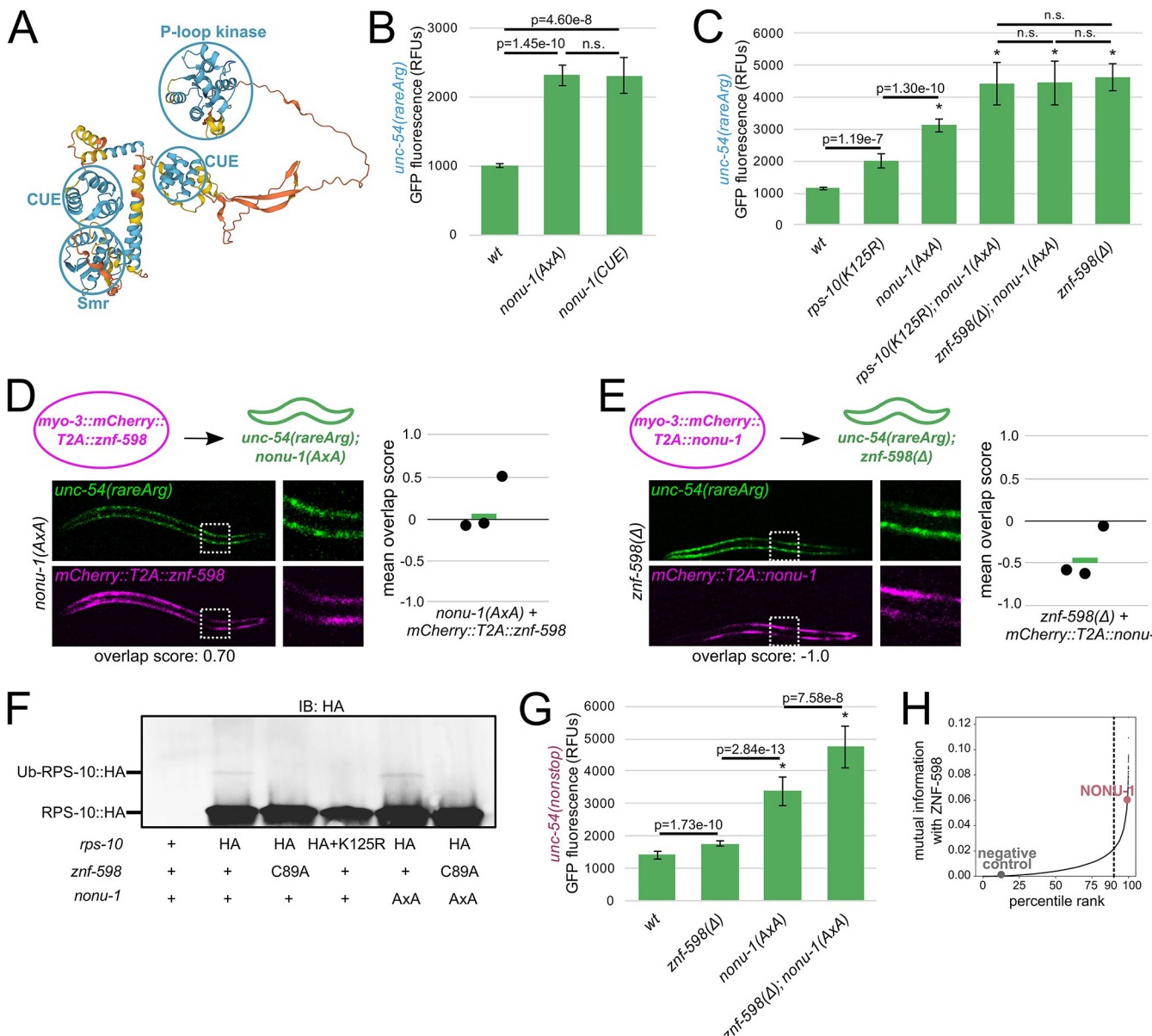

**Fig 5. NONU-1 function during NGD requires CUE domains and follows ZNF-598. (A)** *C. elegans* NONU-1 protein structure as predicted by AlphaFold [29]. Prediction confidence is as follows: blue regions are confident, yellow regions are low confidence, orange regions are very low confidence. Confident domains are circled in blue and labeled. **(B, C)** Mean RFUs (relative fluorescence units) of indicated strains (n≥15 animals/strain) in the *unc-54(rareArg)* background. One standard deviation shown as error bars. p values from Welch's t-test, with asterisks indicating p<0.01 for all comparisons with wild type. **(D)** As in Fig 2, with *znf-598* construct in *nonu-1* strain. **(E)** As in Fig 2, with *nonu-1* construct in *znf-598* strain. **(F)** Western blot of indicated strains to monitor ubiquitination of HA-tagged RPS-10. **(G)** Mean RFUs (relative fluorescence units) of indicated strains (n≥15 animals/strain) in the *unc-54(nonstop)* background. One standard deviation shown as error bars. p values from Welch's t-test, with asterisks indicating p<0.01 for all comparisons with wild type. **(H)** ZNF-598 mutual information plot. 90% percentile cutoff is shown as a dashed line, NONU-1 is highlighted in pink, and a negative control protein is highlighted in gray.

We next investigated the ordering of ZNF-598 and NONU-1 relative to one another in NGD. In a first pair of experiments, we overexpressed ZNF-598 in a *nonu-1* mutant, and also overexpressed NONU-1 in a *znf-598* mutant. According to classical logic when ordering genes in functional pathways, we expect that overexpression of an upstream factor will not compensate for loss of a downstream factor, but that overexpression of a downstream factor will

compensate for loss of an upstream factor [30,31,32,33]. We observed little effect of the *nonu-1* phenotype by ZNF-598 overexpression (Fig 5D), and we saw rescue of the *znf-598* phenotype by NONU-1 overexpression (Fig 5E). These results support a model where NONU-1 acts downstream of ZNF-598. In a second set of experiments, we examined RPS-10 ubiquitination by immunoblot in *nonu-1(AxA)* and observed it to be unchanged (Fig 5F). We interpret this result to indicate that the defect in a *nonu-1* mutant is after ribosomal ubiquitination, *i.e.*, in the commitment of ubiquitinated ribosomes to mRNA decay.

We were curious to determine whether *nonu-1* and *znf-598* function together in NSD as well. Thus, we quantified mutants' effects on the *unc-54(nonstop)* reporter, an allele of *unc-54* tagged with a C-terminal GFP and lacking all stop codons [15,34]. In contrast to its strong effect on the *unc-54(rareArg)* reporter, the *znf-598* mutant showed mild *unc-54(nonstop)* de-repression (Fig 5G). This result is consistent with a more modest role for *znf-598* in NSD than that observed at *unc-54(rareArg)*, and explains why prior NSD screens failed to identify *znf-598* [15,34]. A *nonu-1* mutant exhibited a greater de-repression of *unc-54(nonstop)* than *znf-598*, and the double mutant expressed higher levels of GFP than either single mutant. The fact that *nonu-1* and *znf-598* together exhibit additive effects suggests that NONU-1 is recruited via an E3 ubiquitin ligase other than ZNF-598 during NSD. For more on this, see Discussion.

Taken together, our results with the *unc-54(rareArg)* reporter support a model in which ZNF-598 and NONU-1 function together in NGD. Based on our genetic and molecular analyses, we favor a model in which ribosomal ubiquitination recruits NONU-1 to mRNAs for cleavage.

## A conserved role for ZNF-598 and NONU-1 throughout eukaryotes

To determine if the relationship between ZNF-598 and NONU-1 is a broadly conserved phenomenon throughout eukaryotes, we performed phylogenetic profiling (reviewed in [35]). Briefly, phylogenetic profiling scores pairs of proteins using mutual information, which can be interpreted as the tendency of the protein pair to function together, either redundantly or sequentially. High mutual information can be achieved when two proteins are inherited together or lost together, or when two proteins are genetically redundant and at least one protein is maintained.

We carried out phylogenetic profiling by searching 111,921 profile HMMs [36] on genomic and transcriptomic sequences from 473 protists as these represent diverse eukaryotic lifestyles (Methods). To validate this approach, we calculated mutual information between pairs of factors known to function together in a complex, such as PELO-1/HBS-1 (S4A Fig) and LTN-1/RQC-2 (S4B Fig) [11,12,14,37]. PELO-1/HBS-1 and LTN-1/RQC-2 exhibited high mutual information, scoring in each others' top 99.88% (PELO-1 ranked 19th of 15,909 interactions with HBS-1) and 98.86% (RQC-2 ranked 181th of 15,909 interactions with LTN-1), respectively. Similarly, we observed that ZNF-598 and NONU-1 ranked in each others' top 99.38% (NONU-1 ranked 99th of 15,909 interactions with ZNF-598) (Fig 5H). Thus ZNF-598 and NONU-1 are a broadly conserved functional pair across diverse eukaryotes, with our work in *C. elegans* suggesting that this pair functions in NGD to degrade mRNAs that stall ribosomes.

## HBS-1 N-terminus resembles a ubiquitin-binding domain and is dispensable for NGD

Several of our NGD suppressor mutants (*znf-598*, *nonu-1*, *uba-1*) encoded factors involved in ubiquitin-dependent processes, so we were initially surprised that this screen also identified the ribosome rescue factor *hbs-1*. We therefore examined HBS-1 to determine whether it could conceivably function in a ubiquitin-dependent manner.

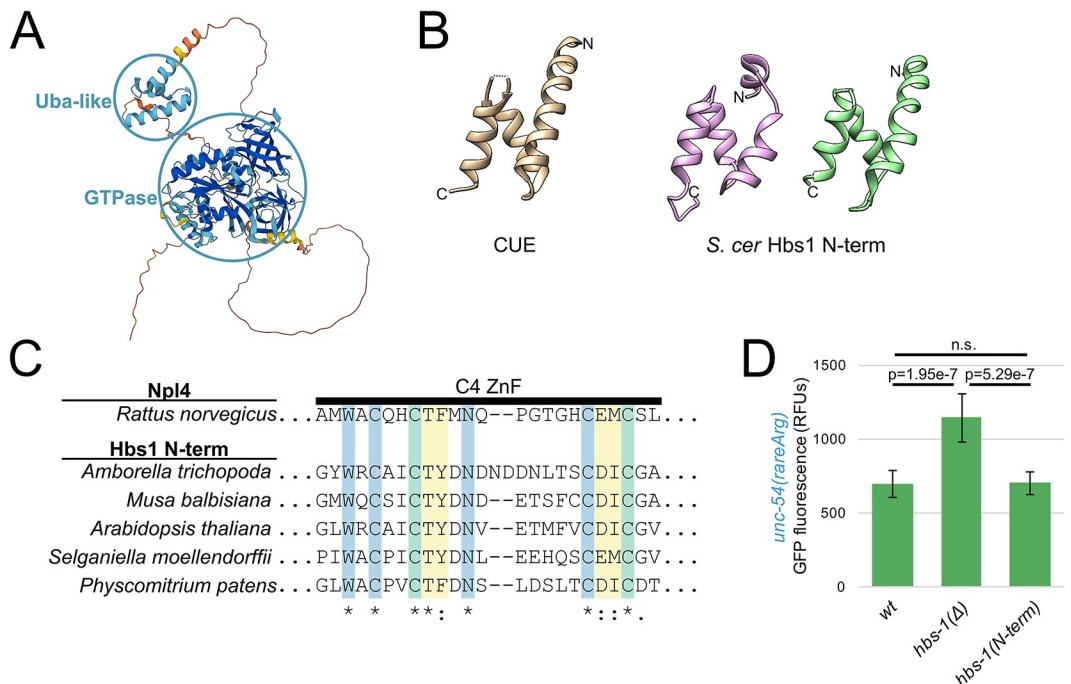

**Fig 6. HBS-1 N-terminus resembles a ubiquitin-binding domain and is dispensable for NGD. (A)** *H. sapiens* Hbs1 protein structure as predicted by AlphaFold [29]. Prediction confidence is as in Fig 5A, with dark blue showing regions of high confidence. Confident domains are circled in blue and labeled. **(B)** Structural homology between Hbs1 N-terminal domain and ubiquitin-binding domains. At left is a structure representative of the UBA clan: CUE from [38] (tan, *S. cerevisiae* Vps9p, 1P3Q). At right are two structures of the Hbs1 N-terminus from [12] (pink, *S. cerevisiae*, 3IZQ) and [14] (green, *S. cerevisiae*, 5M1J). Amino and carboxy termini indicated with N and C, respectively. Note overall similarity in topology and fold across structures. **(C)** The N-terminal zinc finger (ZnF) of plant Hbs1 is homologous to the Ub-binding ZnF of rat Npl4. Multiple sequence alignment of the ZnF of Hbs1 from phylogenetically diverse plants. Residues that are highly conserved among Npl4 homologs are in blue, residues that contact Ub are in yellow, and residues that are both conserved and contact Ub are in green. Coloring and annotation of Npl4 residues from [39]. **(D)** Mean RFUs (relative fluorescence units) of indicated strains (n≥15 animals/strain) in the *unc-54(rareArg)* background. One standard deviation shown as error bars. p values from Welch's t-test.

The HBS-1 protein consists of an N-terminal domain, a disordered linker region, a GTPase domain, and two beta-barrel domains [12,14,29] (Fig 6A). In two published structures, we noticed that the N-terminal domain of HBS-1 is found near known ubiquitination sites on the ribosome (uS10 and uS3) and adopts a distinct triple helix bundle [12,14]. The fold is classified by Pfam as a member of the ubiquitin-binding Uba clan (Fig 6B) [12,14,38]. Notable members of the Uba clan include the Uba and CUE domains, found in *S. cerevisiae* Rad23 and *C. elegans* NONU-1, respectively. We also noticed that plant HBS-1 N-term homologs contain a conserved C4-type zinc finger (ZnF) homologous to a known ubiquitin-binding C4 ZnF domain present in the rat Npl4 (Fig 6C) [39]. These observations show that HBS-1 N-termini from diverse organisms lack sequence or structural homology with each other, and yet many N-termini instead share homology with domains that bind ubiquitin. Given this homology to ubiquitin-binding domains, we hypothesized that the N-terminal domain of HBS-1 may bind ubiquitin during NGD.

To determine whether the N-terminus of HBS-1 is required for its functions in NGD, we generated an N-terminal deletion mutant of *hbs-1* by deleting the region spanning the triple helix bundle and much of the linker region, yielding an HBS-1 consisting of only the first few amino acids, a short linker, and the GTPase domain. Surprisingly, when measuring *unc-54*

*(rareArg)* GFP levels, deleting the N-terminus of HBS-1 had no discernible phenotype, indicating that the N-terminal domain of HBS-1 is dispensable for NGD in *C. elegans* (Fig 6D).

While our genetic analyses supported a role for HBS-1 in NGD independent of the N-terminal domain, it did not rule out a ubiquitin-binding role for the domain outside of NGD. To test this possibility, we performed an *in vitro* ubiquitin-binding assay [40]. However, we failed to observe ubiquitin-binding above background (S5 Fig). We speculate that the success of *in vitro* binding assays may be hindered by a requirement for a ribosome surface to promote HBS-1 binding, as the N-terminus binds to ribosomes in structural studies [12,14]. We look forward to future work to explore the function of the HBS-1 N-terminus on the ribosome.

## HBS-1 and PELO-1 are essential for mRNA degradation

HBS-1 functions with PELO-1 in other contexts, and so we decided to test a functional role of *pelo-1* in NGD. We crossed a *pelo-1* mutant (*cc2849*, bearing a large deletion) [34] into the *unc-54(rareArg)* reporter. The *pelo-1* mutant exhibited a de-repression of *unc-54(rareArg)* comparable to that observed in the *hbs-1* mutant (S6 Fig). Furthermore, the *hbs-1*; *pelo-1* double mutant exhibited a phenotype similar to the single mutants (S6 Fig), suggesting that the two factors function together in NGD as is known in other systems.

HBS-1 and PELO-1 are predominantly known for their role in ribosome rescue, and we were initially surprised by their requirement for repression of the *unc-54(rareArg)* reporter. Despite a well-characterized biochemical role in ribosome rescue [11,12,13,14,41], there is also a known but poorly understood requirement for HBS-1/PELO-1 in mRNA degradation in NGD in other systems [4] as well as in NSD in *C. elegans* [34]. We hypothesized that HBS-1/PELO-1 may facilitate mRNA decay during NGD in *C. elegans*. To test this hypothesis, we performed RNA-seq on *znf-598*, *nonu-1*, and *hbs-1* mutants bearing the *unc-54(rareArg)* reporter (Fig 7A). We observed an increase in *unc-54(rareArg)* mRNA levels comparable to the level of GFP de-repression observed in each strain, with *znf-598* conferring the highest levels and *nonu-1* and *hbs-1* exhibiting a lesser and similar increase in mRNA levels. Therefore, we conclude that HBS-1 is required for mRNA degradation during NGD in *C. elegans*.

Studies in *S. cerevisiae* identified homologs of *hbs-1* and *pelo-1* (*HBS1* and *DOM34*) as being required for endonucleolytic cleavages during NGD [4,10,42], though at the time of that work, the identity of the nuclease was unknown. Based on this literature and our RNA-seq analysis, we hypothesized that HBS-1/PELO-1 may act in the same pathway as NONU-1. To test this hypothesis, we performed double mutant analysis with *hbs-1* and *nonu-1*. The *nonu-1*; *hbs-1* double mutant was indistinguishable from a *nonu-1* single mutant, suggesting that HBS-1 acts in the same pathway as NONU-1 (Fig 7B). We also found HBS-1 to act in the same pathway as ZNF-598 (Fig 7C).

To place HBS-1 relative to ZNF-598 and NONU-1 in NGD, we tested whether excess ZNF-598 or NONU-1 could compensate for the loss of *pelo-1* and *hbs-1*. Overexpression of ZNF-598 resulted in little change to the NGD phenotype (Fig 7D), consistent with a model where HBS-1/PELO-1 act downstream of ZNF-598. In contrast, overexpression of NONU-1 rescued NGD in *pelo-1*; *hbs-1* animals (Fig 7E). This result suggests that NONU-1 acts downstream of HBS-1/PELO-1. Interestingly, analyses using the NSD reporter indicate separate capabilities for NONU-1 and HBS-1/PELO-1 in NSD: a double mutant analysis of *pelo-1* and *nonu-1* in NSD displayed an additive effect (Fig 7F).

Taken together, our analyses place HBS-1 and PELO-1 on a pathway with ZNF-598 and NONU-1 during NGD, and suggests a model where HBS-1 acts downstream of ZNF-598 to promote mRNA decay by NONU-1 (Fig 7G).

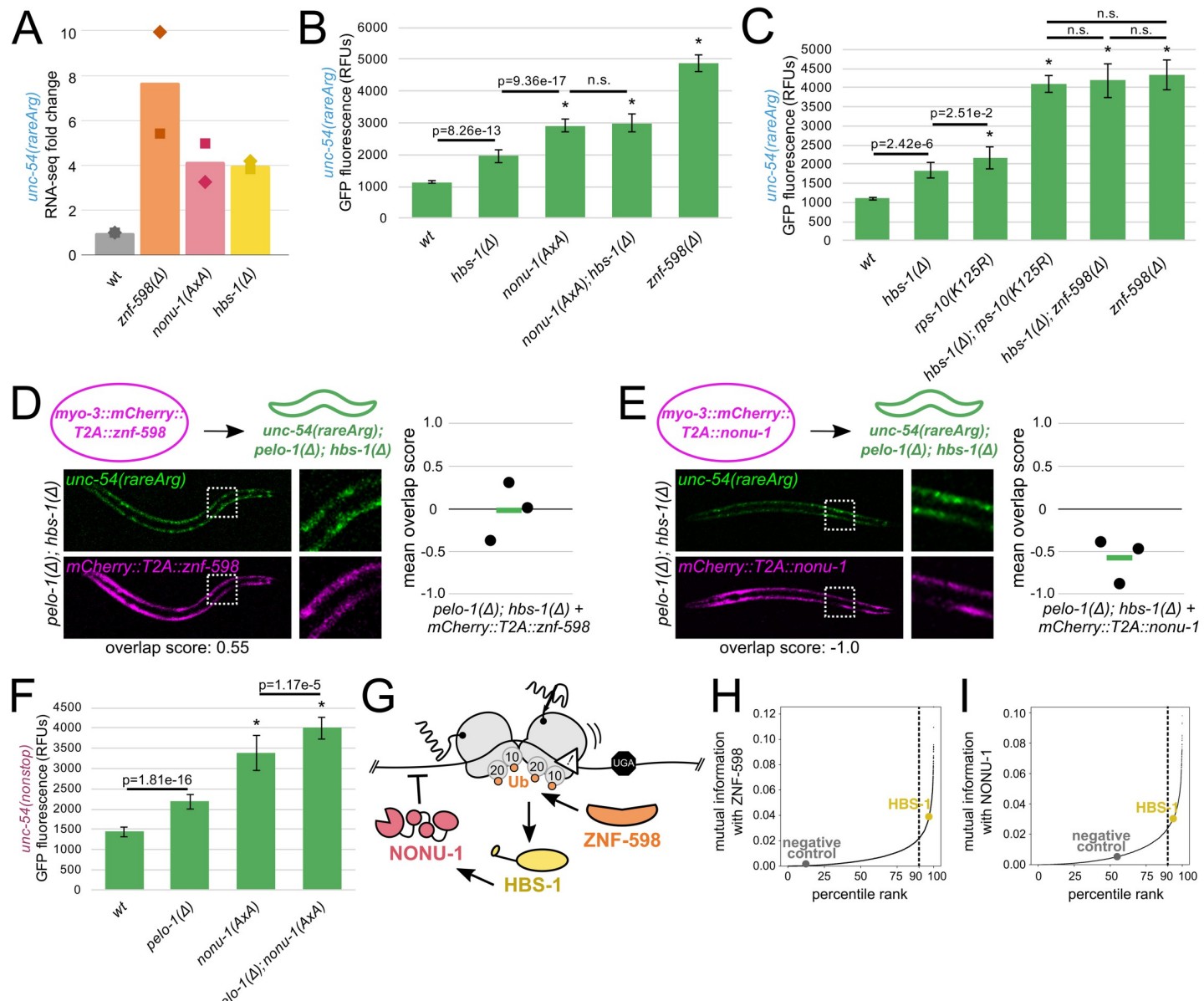

**Fig 7. HBS-1 and PELO-1 are essential for mRNA degradation. (A)** RNA-seq mean fold change of *unc-54(rareArg)* in the indicated strains from two biological replicates (shown as diamonds and squares). **(B, C)** Mean RFUs (relative fluorescence units) of indicated strains (n≥15 animals/strain) in the *unc-54(rareArg)* background. One standard deviation shown as error bars. p values from Welch's t-test, with asterisks indicating p<0.01 for all comparisons with wild type. **(D)** As in Fig 2, with *znf-598* construct in *pelo-1; hbs-1* strain. **(E)** As in Fig 2, with *nonu-1* construct in *pelo-1; hbs-1* strain. **(F)** Mean RFUs (relative fluorescence units) of indicated strains (n≥15 animals/strain) in the *unc-54(nonstop)* background. One standard deviation shown as error bars. p values from Welch's t-test, with asterisks indicating p<0.01 for all comparisons with wild type. **(G)** Model for NGD via ZNF-598, HBS-1, and NONU-1. **(H)** ZNF-598 mutual information plot. 90% percentile cutoff is shown as a dashed line, HBS-1 is highlighted in yellow, and a negative control protein is highlighted in gray. **(I)** As in (H), showing NONU-1 mutual information with HBS-1.

## HBS-1 is broadly conserved with NONU-1 and ZNF-598

In light of our genetic analyses, we were curious to measure the conservation of the relationships between HBS-1 and both NONU-1 and ZNF-598. To this end, we carried out phylogenetic profiling of HBS-1 with ZNF-598 homologs, and observed a relatively high level of mutual information falling in each others' 96.9%/98.3% (Fig 7H), suggesting that HBS-1 and

ZNF-598 function together throughout eukarya. NONU-1 and HBS-1 fell in each others' 93.7% and 96.7%, still exhibiting some conservation, but less so than either factor with ZNF-598 (Fig 7I). Taken together, our conservation analyses support a conserved relationship between ZNF-598 and HBS-1, with a less conserved relationship between HBS-1 and NONU-1.

## Discussion

Here, we investigated the functions of and interplay between several factors required for NGD. Overall, our work supports a model where ubiquitination of RPS-10 and RPS-20 by ZNF-598 recruits NONU-1 via ubiquitin-binding to elicit mRNA decay, and HBS-1 acts downstream of ZNF-598 to enable mRNA cleavage by NONU-1.

Our data are consistent with a model in which ZNF-598 directly ubiquitinates RPS-10 and RPS-20. We note that our data could also be explained by a model in which ZNF-598 indirectly affects RPS-10 and RPS-20 ubiquitination, *e.g.*, through an intermediary E3 ligase. While we did not identify other E3 ligases in our screen, it is possible that loss of such E3 ligases would be inviable, precluding their recovery. Additional work will be required to distinguish between these models.

Regardless of whether ZNF-598 acts directly or indirectly on RPS-10 and RPS-20, our data suggest these proteins act in a single pathway to bring about mRNA repression. There may also be additional ubiquitination sites functioning on the ribosome during NGD, as has been suggested in other systems [43,44]. It is notable that we can efficiently block mRNA degradation via ablation of RPS-10 and RPS-20 sites alone. Our analysis of NONU-1's relationship to ribosomal ubiquitination sites suggests a model that is more complicated than a simple one-ubiquitination-site-one-effector model. This may be expected given NONU-1's multiple ubiquitin-binding domains. Further genetic work will be required to elucidate the network of ubiquitination sites functioning in NGD, and given the combinatorial possibilities of sites on collided ribosomes, we expect structural studies to be insightful.

While our *unc-54(rareArg)* reporter indicated that NONU-1 and ZNF-598 largely function in the same pathway during NGD, our NSD data are consistent with flexibility in NONU-1's recruitment. Given the minor effect of ZNF-598 in NSD, and the absence of *znf-598* alleles in NSD screens [15,34], we hypothesize that a factor other than ZNF-598 is used to recognize ribosomes stalled at the 3'end of an mRNA. Other E3 ligases have been implicated in surveillance, and future work will hopefully clarify the recruitment mechanisms of each [7,44,45]. These data could also be explained by differences in the stalling mechanism and/or reporter readout on *unc-54(nonstop)* vs. *unc-54(rareArg)*. Regardless, the mild NSD de-repression seen in *znf-598* indicates that ZNF-598 does act on some ribosomes during NSD. We hypothesize that collisions targeted by ZNF-598 are worsened in a *nonu-1* mutant, explaining the greater effect of *znf-598* on NSD in a *nonu-1* mutant (Fig 5G). These ideas are consistent with other models suggesting multiple types of collisions, with the relative amount of each depending on the genetic background [7].

Here, we discovered that the N-terminal domain of HBS-1 is dispensable for NGD, despite sharing structural homology to ubiquitin-binding domains. Our sequence analyses revealed that HBS-1 N-termini encode domains homologous to ubiquitin-binding domains despite little primary sequence conservation. While this suggests that HBS-1 binds ubiquitin, we failed to observe ubiquitin binding above background, suggesting that additional factors may be required for an interaction. It is also possible that the HBS-1 N-terminus interferes in the ubiquitination reaction. Given that current structures of ribosome-bound HBS-1 lack density for ubiquitin, we expect structural studies with ubiquitinated ribosomes will prove insightful.

Given prior work in *S. cerevisiae* revealing a role for Hbs1 in NGD cleavage [4,10,42], and in light of our data here, we favor a model where HBS-1 activity greatly enhances mRNA

cleavage by NONU-1. We note that we recovered an HBS-1 GTPase mutant in our screen, suggesting that GTP hydrolysis is required for HBS-1 function in mRNA decay. We therefore hypothesize that, in the process of rescuing ribosomes, HBS-1/PELO-1 generate a substrate for NONU-1 or otherwise enable NONU-1 action. In this model it is unclear which ribosomes are rescued, but prior work supports HBS-1/PELO-1 function on internally-stalled ribosomes [41], consistent with a rescue-first-cleavage-second NGD model. These mechanistic ties between HBS-1 and NONU-1 will enable discovery of the molecular mechanisms underlying the known but poorly understood link between ribosome rescue and mRNA decay.

Overall, our work augments information gained from other systems by demonstrating the conservation of NGD factors in a metazoan, and ordering the sequence of events carried out by ZNF-598, HBS-1/PELO-1, and NONU-1 to bring about mRNA repression in response to ribosomal stalls.

## Methods

### *C. elegans* strain construction and propagation

*C. elegans* strains were derived from theVC2010 strain (N2) [46], and the CB4856 (Hawaiian) strain was used for suppressor mutant mapping. Animals were grown at 16C or 20C on NGM plates seeded with OP50 as a food source [47]. Some strains were obtained from the Caenorhabditis Genetic Center (CGC), which is funded by NIH Office of Research Infrastructure Programs (P40 OD010440). CRISPR/Cas9 was performed to introduce genomic edits as previously described [48]. Multiple independent isolates of each mutation were recovered and observed to have identical phenotypes. Mutant combinations were generated by crossing. A full list of strains, sequences of mutant alleles, PCR primers, and sources is available in S1 Table.

### EMS mutagenesis and suppressor mutant screens

EMS mutagenesis was performed essentially as described [34]. Briefly, a large population of each strain was washed with M9 to a final volume of 4mL. EMS was added to a final concentration of ~1mM and animals were incubated at room temperature for 4 hr with nutation. Animals were washed and left overnight at room temperature on NGM plates seeded with OP50. The next day, animals were washed and eggs were collected by hypochlorite treatment. 100–200 eggs were placed on single small NGM+OP50 plates and allowed to propagate. Plates were screened for F2/F3 animals with increased GFP fluorescence and increased movement. Only a single isolate was kept per NGM plate to ensure independence of mutations.

The first rareArg screen was saturated with hits at *znf-598*, occluding our ability to recover additional alleles at other loci. To address this, we constructed a double-balanced strain covering the region of chromosome II harboring *znf-598*, similarly to an approach used in an NMD suppressor screen [49]. This strain (WJA 1040) was homozygous on chromosome I for *unc-54 (srf1004)* and heterozygous on chromosome II for *mnC1[dpy-10(e128) unc-52(e444) nls190 let-?]* (AG226) and *tra-2(e1095)* (CB2754). Animals homozygous for *mnC1* or homozygous for *tra-2(e1095)* were inviable or Tra/sterile, respectively, allowing us to eliminate recovery of recessive suppressor mutants within the balanced region. This strain was subjected to EMS mutagenesis as described above. Only isolates maintaining the *mnC1/tra-2* heterozygous balancer were kept, as seen by *myo-2*::*GFP* (marking *mnC1*), *a* nonTra phenotype, and subsequent viability.

### Suppressor mutant mapping and identification

Following recovery of mutants from EMS screens, we employed a Hawaiian SNP mapping approach as described [50]. We crossed each isolated suppressor mutant to Hawaiian

(CB4856) *unc-54(cc4112)* males (expressing an UNC-54::mCherry fusion engineered by CRISPR/Cas9). Cross progeny were isolated and allowed to self-fertilize. The F2 GFP+ progeny were then backcrossed to the *unc-54(cc4112)* males at least 2 times.

After rounds of backcrossing, several phenotypically suppressed animals (~20–30) for a given mutant were pooled together onto a single plate and propagated until starvation. Upon starvation, animals were washed off the plate with 1mL EN50, and further washed with EN50 to remove residual *E. coli*. Genomic DNA was extracted after proteinase K treatment and resuspended in 50uL TE pH7.4. 50ng genomic DNA was used as an input for Nextera (Tn5) sequencing library preparation. Libraries were sequenced at Novogene Corporation Inc. UC Davis Sequencing Center on a NovaSeq 6000.

Reads were mapped to the *C. elegans* genome using bowtie2 (version 2.3.4.1). Reads were assigned to haplotypes using GATK [51] and a previously published dataset of Hawaiian SNPs [46]. The fraction of reads that were assignable to Hawaiian or N2 animals was calculated across the genome, and linkage was identified by portions of the genome with 0% Hawaiian. Regions of linkage were then manually inspected to identify candidate lesions and loci.

### Fluorescence microscopy and image analysis

All animals were maintained at 20C. L4 animals were selected and anesthetized in 3uL EN50 with 1mM levamisole in a microscope well slide with a 0.15mm coverslip.

A Zeiss AxioZoom microscope was used with a 1.0x objective to acquire images for GFP quantification experiments. The following parameters were used for all images: exposure time of 250ms., shift of 50%, and zoom of 80%. 15–25 representative animals were imaged for each strain. All comparisons shown are between images obtained during the same imaging session. We used FIJI to define the area of the animal, subtract the background, and determine mean pixel intensity for the area of each animal.

A Leica SP5 confocal microscope was used with a 10.0x objective to simultaneously acquire images of mCherry and GFP in overexpression experiments. The following parameters were used for all images: pinhole size set to 106.2 μm; bidirectional scan at 400 Hz with a line average of 2; 488 laser to capture GFP and 594 laser to capture mCherry, both at 50% power with a gain of 700 and offset of -0.2%. 4–10 animals were analyzed from a total of 3 independent isolates for each overexpression experiment. A custom python script was used to define the area of the animal, then parse out locations and intensities of green and red pixels. We considered the brightest 1,000 green and brightest 1,000 red pixels and calculated the Jaccard index. The theoretical maximum of the Jaccard index is 1, indicating no overlap of green and red pixels. The observed minimum of the Jaccard index among our strains was ~0.75, higher than the theoretical lower bound of zero, which is the result of expression of mCherry and GFP being driven in the same tissue. To display the image quantification more intuitively, we transformed the Jaccard index into an overlap score. This was done by performing a simple linear transformation of 0.75 (more overlap) to 1.0 (less overlap) in Jaccard space into an overlap score of -1.0 (less overlap) to 1.0 (more overlap). See S1 Fig for images of animals representing various overlap scores.

### Immunoblotting

Animals were propagated on NGM plates with OP50 until freshly starved, then washed twice with 1ml M9 and flash frozen in liquid nitrogen. Animal pellets were boiled for 10 min at 99C in 1x SDS loading buffer (0.1M Tris-HCl, 20% glycerol, 4% SDS, 0.1M DTT, 0.05% bromophenol blue). Samples were vortexed and spun to pellet animals and supernatants were collected. Protein was quantified by Qubit and 15ug protein was run on a 4–20% Mini-PROTEAN(R)

TGX Stain-Free Protein gel (Bio-Rad). Protein was transferred to a low background fluorescence PVDF membrane (Millipore Sigma) and the membrane was blocked in 5% nonfat milk in 1x TBST. Anti-HA antibody (Enzo Life Sciences 16B12) was used at a 1:2000 dilution to detect HA-tagged RPS-10 protein. Anti-FLAG antibody (Millipore Sigma F1804) was used at a 1:1000 dilution to detect FLAG-tagged RPS-20 protein. Anti-H3 antibody (Millipore Sigma SAB4500352) was used at a 1:2000 dilution to detect the histone H3 protein. Secondary antibody staining was performed with 1:15000 LI-COR goat anti-mouse or LI-COR goat anti-rabbit (LI-COR). Imaging was done using a LI-COR Odyssey Imaging System (LI-COR), with quantification done in ImageStudio.

### Mutual information calculation

Phylogenetic profiling was performed using the PANTHER HMM library Version 16.0 on Ensembl Protists Genome (release 51) combined with MMETSP [52]. To identify homologs of proteins from the PANTHER Subfamily HMMs, we used HMMER's hmmsearch function using the following parameters—noali—notextw—cpu 2. We then parsed each output file from HMMER and determined the presence (1) or absence (0) of a homolog in each organism for a sub-family HMM.

Briefly, if multiple subfamily HMMs matched a given protein in a species, the protein was assigned to the subfamily with the highest bit score, and lesser-scoring subfamilies were ignored. Subfamilies with homologs in <5% or >95% of species were discarded; such subfamilies exhibit too little variation for accurate mutual information calculations. Similarly, species with hits for <10% of subfamilies were discarded; such species may represent poor transcriptome coverage and/or assemblies. We also computed the pairwise hamming distance between all species and discarded species until the pairwise hamming distance between all species was at least 0.1, so as to ensure biodiversity. Mutual information for discrete variables was then calculated as: $\text{sum}(i = 0,1); \text{sum}(j = 0,1) [-\log_2(p\_ij/(p\_i^*p\_j))]$.

Negative controls for mutual information calculations were generated as follows. For a given pair of factors A and B, we randomized the binary vector of presence (1) and absence (0) of B in organisms to create a new negative control factor. Mutual information for discrete variables was then calculated between factor A and the newly created negative control.

### RNA-seq and analysis

Animals were synchronized by hypochlorite treatment, propagated on NGM plates with OP50 at 16C, and harvested at the L4/young adult stage. Animals were washed off with N50, passed through a 5% sucrose cushion in N50 to remove *E. coli*, and snap frozen in liquid nitrogen. Animals were lysed by grinding in a mortar and pestle cooled in liquid nitrogen in the presence of frozen PLB (20mM Tris pH8.0, 140mM KCl, 1.5mM MgCl2, 1% Triton) and 100ug/mL cycloheximide. Ground animals were stored as frozen powder at -70C. Total RNA was extracted with trizol, resuspended in TE pH7.4, and quantified by Qubit. Ribosomal RNA was depleted using custom *C. elegans*-specific rRNA hybridization oligos, similar to a planarian protocol as described [53]. Oligos for this protocol are included in S1 Table. Libraries were prepared using an NEBNext Ultra II Directional RNA Library Prep Kit for Illumina sequencing. Libraries were sequenced at Novogene Corporation Inc. UC Davis Sequencing Center on a NovaSeq 6000 and at the Vincent J. Coates Genomics Sequencing Laboratory at UC Berkeley on a HiSeq 4000.

We generated a custom *C. elegans* genome (Ensembl, WBCel235) containing *unc-54(rareArg)* as a separate chromosome and a masked endogenous *unc-54* locus. Reads were mapped to this genome including annotated splice junctions using STAR v2.5.4b [54] allowing for six mismatches. All downstream analyses were restricted to uniquely mapping reads.

### *In vitro* ubiquitin-binding

Ubiquitin-binding assay was performed essentially as described in [40]. Briefly, recombinant proteins were induced in *E. coli* with 1 mM IPTG at 37C for 4 hours. Cells were harvested and suspended in 1x PBS (137mM NaCl, 2.7mM KCl, 10mM Na2HPO4, and 1.8mM KH2PO4), followed by lysis in xTractor buffer (Takara) per the manufacturer's protocol. Lysates were cleared by centrifugation and immobilized on TALON metal affinity resin (Takara) per the manufacturer's protocol. Immobilized proteins were washed with 1x PBS and incubated overnight at 4C with clarified lysates containing various GST-fusion constructs. Bound proteins were washed four times with 1x PBS with 5mM imidazole and eluted by heating to 99C for 5 min in 1x SDS loading buffer. Eluted protein was run on a 4–20% Mini-PROTEAN(R) TGX Stain-Free Protein gel (Bio-Rad) and stained with Coomassie brilliant blue.

## Supporting information

**S1 Table. *C. elegans* strains and oligos.**
(XLSX)

**S2 Table. Raw numerical values for all graphs.**
(XLSX)

**S1 Fig. Representative animals demonstrating various overlap scores.** GFP and mCherry images of representative animals expressing *unc-54(rareArg)* and an mCherry-tagged array. Above is the calculated overlap score.
(TIF)

**S2 Fig. RPS-10 and RPS-20 ubiquitination sites are conserved in *C. elegans*. (A)** Pairwise sequence alignment of *H. sapiens* eS10 and *C. elegans* RPS-10. K125 is highlighted in green. Conservation is as shown in Fig 3A. **(B)** As in (A), showing *H. sapiens* uS10 and *C. elegans* RPS-20 with K6 and K9 highlighted in green.
(TIF)

**S3 Fig. *nonu-1* null mutant displays similar NGD suppression as catalytic mutant.** Mean RFUs (relative fluorescence units) of indicated strains (n≥15 animals/strain) in the *unc-54 (rareArg)* background. One standard deviation shown as error bars. p values from Welch's t-test, with asterisks indicating p<0.01 for all comparisons with wild type.
(TIF)

**S4 Fig. Known surveillance factor partners exhibit high mutual information. (A)** HBS-1 mutual information plot. 90% percentile cutoff is shown as a dashed line and PELO-1 is highlighted in purple. **(B)** As in (A), showing LTN-1 mutual information with RQC-2 in red.
(TIF)

**S5 Fig. Ubiquitin-binding assay shows high level of background binding.** His6-tagged constructs of *S. cerevisiae* Cue2 CUE domains, *H. sapiens* HBS1 N-term domain, *H. sapiens* HBS1 N-term globular (triple helix) domain, *H. sapiens* HBS1 N-term globular (triple helix) domain with G93D mutation (Gly in predicted binding site found from [55]), and *H. sapiens* SMG5 PIN domain. His6 constructs were immobilized on cobalt metal affinity resin and incubated with *E. coli* lysates expressing GST-Ub or GST. Proteins boiled from resin are shown on Coomassie stained gel, with proteins indicated. Asterisk indicates nonspecific peptide present on resin.
(TIF)

**S6 Fig. HBS-1 and PELO-1 exhibit similar phenotypes during NGD in *C. elegans*.** Mean RFUs (relative fluorescence units) of indicated strains (n≥15 animals/strain) in the *unc-54 (rareArg)* background. One standard deviation shown as error bars. p values from Welch's t-test, with asterisks indicating p<0.01 for all comparisons with wild type.
(TIF)

## Acknowledgments

We thank members of the Arribere lab for thoughtful discussions. We thank Benjamin Abrams at the UCSC Life Sciences Microscopy Center for technical support (RRID: SCR_021135). We thank Wormbase.

## Author Contributions

**Conceptualization:** Parissa C. Monem, Joshua A. Arribere.

**Formal analysis:** Parissa C. Monem, Joshua A. Arribere.

**Funding acquisition:** Joshua A. Arribere.

**Investigation:** Parissa C. Monem, Nitin Vidyasagar, Audrey L. Piatt, Enisha Sehgal, Joshua A. Arribere.

**Supervision:** Joshua A. Arribere.

**Visualization:** Parissa C. Monem, Joshua A. Arribere.

**Writing – original draft:** Parissa C. Monem, Joshua A. Arribere.

**Writing – review & editing:** Parissa C. Monem, Joshua A. Arribere.

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
