## [Decision Letter · Decision Letter 0]

13 Dec 2022

Dear Dr Arribere,

Thank you very much for submitting your Research Article entitled 'Ubiquitination of stalled ribosomes enables mRNA decay via HBS-1 and NONU-1 in vivo' to PLOS Genetics.

The manuscript was fully evaluated at the editorial level and by independent two peer reviewers. The reviewers are essentially ready to recommend acceptance, but Reviewer #2 has a handful of minor, easily addressable, but nonetheless important issues.  After you attend to these I will be able to proceed with an editorial decision without further external evaluation.

We therefore ask you to modify the manuscript according to the review recommendations. Your revisions should address the specific points made by each reviewer.

Yours sincerely,

Gregory P. Copenhaver

Editor-in-Chief

PLOS Genetics

Reviewer's Responses to Questions

**Comments to the Authors:**

Reviewer #1: The authors successfully addressed my remaining concerns and the manuscript has been greatly improved.

Reviewer #2: This is a very thoroughly revised manuscript that is greatly improved by the addition of new data, removal of data irrelevant to the main point, and by thoroughly revising the text. In this manuscript the authors study no-go mRNA decay in C. elegans. They very effectively use the powerful genetics of this system to identify factors required for no-go decay and then follow up with experiments that provide strong evidence that supports the main conclusion that ribosomal ubiquitination enables mRNA degradation via HBS-1 and NONU-1 in an intact animal. The revised manuscript is a valuable extension of what is known and should be published in PLOS Genetics. I have some very minor comments for the authors that may help them further improve the manuscript detailed below

Minor comments:

1. The authors isolated 20 mutants but only describe 9 alleles. If he authors identified mutations in the other 11 mutants, it would be helpful to the field to add those to table S1, for example if another researcher identifies mutations in the same genes or their orthologs. If the other 11 alleles were not mapped and sequenced, perhaps the authors can clarify that.

2. In line 298 the authors mention creating tagged alleles of NONU-1 that turned out to be nonfunctional. It might be useful to the field to include the details in the supplement. This would prevent future researchers from wasting time trying the same tagging strategy.

3. The use of “phenocopy” in lines 223, 285, 307, 308, and 427 is at best unnecessary jargon but according to the definition of phenocopy in my dictionary or in Wikipedia is actually incorrect. Phenocopy should mean that some specific treatment (e.g. a drug) has the same effect as a mutation and not that two mutations have the same effect. The authors correctly describe mutations with similar phenotypes in lines 296 and 403-406. Even worse is in line 389 where a mutant is described as phenocopying wild type. Much clearer would be to say that deleting the N-terminus of HBS-1 had no discernible phenotype.

4. Line 189 “containing fluorescent protein fusions” is incorrect. The T2A stop and go sequence means that mCherry and ZNF-598 (or NONU-1) are produced as separate proteins and not fusion proteins. In addition, plasmids may encode proteins, but do not “contain” them.

5. Line 429 “acts” should be “act”

**Have all data underlying the figures and results presented in the manuscript been provided?**

Reviewer #1: Yes

Reviewer #2: Yes

PLOS authors have the option to publish the peer review history of their article (what does this mean?). If published, this will include your full peer review and any attached files.

Reviewer #1: **Yes: **Seung-Jae V. Lee

Reviewer #2: No

---

## [Editor Report · Decision Letter 1]

18 Dec 2022

Dear Dr Arribere,

We are pleased to inform you that your manuscript entitled "Ubiquitination of stalled ribosomes enables mRNA decay via HBS-1 and NONU-1 in vivo" has been editorially accepted for publication in PLOS Genetics. Congratulations!

Yours sincerely,

Gregory P. Copenhaver

Editor-in-Chief

PLOS Genetics

Comments from the reviewers (if applicable):

**Data Deposition**

http://datadryad.org/submit?journalID=pgenetics&manu=PGENETICS-D-22-01367R1

**Press Queries**

---

## [Editor Report · Acceptance letter]

5 Jan 2023

PGENETICS-D-22-01367R1 

Ubiquitination of stalled ribosomes enables mRNA decay via HBS-1 and NONU-1 in vivo 

Dear Dr Arribere, 

We are pleased to inform you that your manuscript entitled "Ubiquitination of stalled ribosomes enables mRNA decay via HBS-1 and NONU-1 in vivo" has been formally accepted for publication in PLOS Genetics! Your manuscript is now with our production department and you will be notified of the publication date in due course.

With kind regards,

Zsuzsanna Gémesi

PLOS Genetics

On behalf of:
